# Named Entity Recognition
# as Graph Classification

Ismail Harrando[✉] and Raphaël Troncy

EURECOM, Sophia Antipolis, France
{ismail.harrando,raphael.troncy}@eurecom.fr

**Abstract.** Injecting real-world information (typically contained in Knowledge Graphs) and human expertise into an end-to-end training pipeline for Natural Language Processing models is an open challenge. In this preliminary work, we propose to approach the task of Named Entity Recognition, which is traditionally viewed as a *Sequence Labeling* problem, as a *Graph Classification* problem, where every word is represented as a node in a graph. This allows to embed contextual information as well as other external knowledge relevant to each token, such as gazetteer mentions, morphological form, and linguistic tags. We experiment with a variety of graph modeling techniques to represent words, their contexts, and external knowledge, and we evaluate our approach on the standard CoNLL-2003 dataset. We obtained promising results when integrating external knowledge through the use of graph representation in comparison to the dominant end-to-end training paradigm.

**Keywords:** Named Entity Recognition · Graph Classification.

## 1  Introduction

Transformer-based language models such as BERT [2] have tremendously improved the state of the art on a variety of Natural Language Processing tasks and beyond. While it is hard to argue against the performance of these language models, taking them for granted as the fundamental building-block for any NLP application stifles the horizon of finding new and interesting methods and approaches to tackle quite an otherwise diverse set of unique challenges related to specific tasks. This is especially relevant for tasks that are known to be dependent on real-world knowledge or domain-specific and task-specific expertise. Although these pre-trained language models have been shown to internally encode some real-world knowledge (by virtue of being trained on large and encyclopedic corpora such as Wikipedia), it is less clear which information is actually learnt and how it is internalized, or how one can inject new external information (e.g. from a knowledge base) into these models in a way that it does not require retraining them from scratch.

In this work, we propose a novel method to tackle Named Entity Recognition, a task that has the particularity of relying on both the linguistic understanding of the sentence as well as some form of real-world information, as what makes a

Named Entity is the fact that it refers to an entity that is generally designated by a proper name. Since graphs are one of the most generic structures to formally represent knowledge (e.g. Knowledge Graphs), they constitute a promising representation to model both the linguistic (arbitrarily long) context of a word as well as any external knowledge that is deemed relevant for the task to perform. Graph connections between words and their descriptions seems to intuitively resemble how humans interpret words in a sentence context (how they relate to preceding and following words, and how they relate external memorized knowledge such as being a "city name" or "an adjective"). Hence, we propose to cast Named Entity Recognition as a Graph Classification task, where the input of our model is the representation of a graph that contains the word to classify, its context, and other external knowledge modeled either as nodes themselves or as node features. The output of the classification is a label corresponding to the entity type of the word (Figure 1).

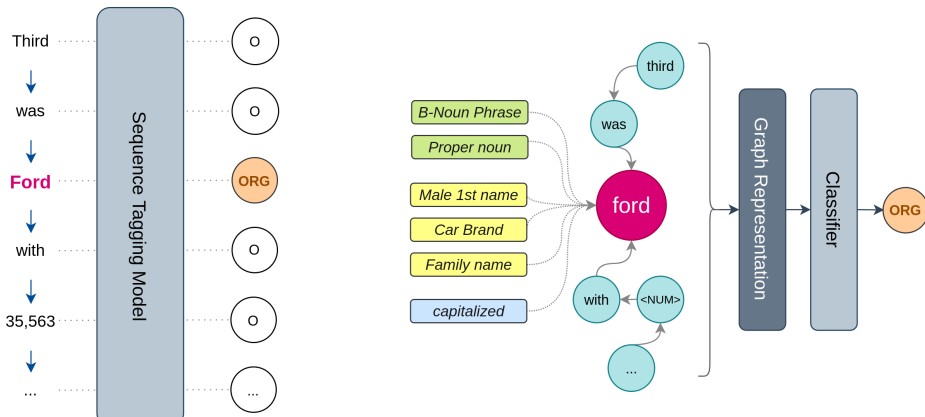

**Fig. 1.** Left: Traditional sequence tagging model. Right: Each *word* in a sentence becomes the central node of a graph, linked to the words from its context, as well as other task-related features such as grammatical properties (e.g. "Proper Noun"), gazetteers mentions (e.g. "Car Brand") and task-specific features (e.g. "Capitalized"). The graph is then embedded which is passed to a classifier to predict an entity type.

## 2   Approach

In order to perform Named Entity Recognition as a Graph Classification task, the "word graph" needs to be transformed into a fixed-length vector representation, that is then fed to a classifier, e.g. a feed-forward neural network (see figure 1). This graph representation needs to embed the word to classify (the *central node*), as well as its *context* – words appearing before and after it – and its related *tags* (properties such as gazetteers mention, grammatical role, etc). This formalization is interesting because it allows to represent the entire context of the word (as graphs can be arbitrarily big), to explicitly model the left and the right context

separately, and to embed different descriptors to each word seamlessly (either as node features or as other nodes in the graph) and thus help the model to leverage knowledge from outside the sentence and the closed training process. This is a first difference with the traditional sequence labeling methods that only consider a narrow window the tokens to annotate. While we posit that this method can integrate any external data in the form of new nodes or node features in the input graph, we focus on the following properties that are known to be related to the NER task:

- **Context**: which is made of the words around the word we want to classify.
- **Grammatical tags**: we use the Part of Speech tags (`POS`) e.g. 'Noun', as well as the shallow parsing tags (`chunking`) e.g. 'Verbal Phrase'.
- **Case**: in English, capitalization is an important marker for entities. We thus add tags such as: 'Capitalized' if the word starts with a capital letter, 'All Caps' if the word is made of only uppercase letters, and so on.
- **Gazetteers**: we generate lists of words that are related to potential entity types by querying Wikidata for labels and synonyms corresponding to entities belonging to types of interest such as *Family Name*, *Brand*, etc.

The literature on Graph Representations shows a rich diversity in approaches [1, 4], but for our early experiments, we choose one candidate from each of the main representation families: a neural auto-encoder baseline, Node2Vec for node embeddings, TransE for Entity Embeddings, and a Graph Convolutional Network based on [4]. This is admittedly a small sample of the richness that can be further explored in the future, both in terms of the models and the way the input graph is constructed (how to model the context and the added knowledge).

## 3   Experiments and Results

### 3.1   Experimental protocol

To train each of the aforementioned models, we construct a dataset[1] by going through every word in every document from the CoNLL training dataset, and build its graph (Figure 1). Each of these graphs is then turned into a fixed-length vector that is fed to a neural classifier (Section 3.2). For each of the representations, we fine-tune the hyper-parameters using the ConLL validation (dev) set. We report the Micro-F1 and Macro-F1 scores for all trained models in Table 1 for both the validation and the test sets together with the best currently reported performance approach from the state of the art[2].

### 3.2   Methods

To evaluate the approach, we selected the following methods to generate graph embeddings:

---

[1] `https://github.com/Siliam/graph_ner/tree/main/dataset/conll`

[2] See also `http://nlpprogress.com/english/named_entity_recognition.html`

1. `Binary Auto-encoder`: we represent the word graph as a binary vector. We concatenate one-hot embeddings of the word, its left and right context, and all other extra tags in the vocabulary (e.g. POS tags, gazetteers mention, etc.). We use this "flat" representation of the graph as a baseline that incorporates all the external data without leveraging the graph structure. We first train a neural encoder-decoder (both feed-forward neural networks with one hidden layer) to reconstruct the input binary representation of the graph. We then use the encoder part to generate a graph embedding to feed to our final classifier.

2. `Node2Vec`: we generate the graph representing all nodes in the training set (all words as related to their context, with the external knowledge tags also represented as nodes), and then we use `Node2Vec`[3] to generate embeddings for all nodes. The final input graph representation is obtained by averaging all nodes representations, i.e. the word, its context and its tags.

3. `TransE`: we generate the graph as with the `Node2Vec` method, except that the edges between the different nodes (entities) are now labeled relations such as 'before', 'after', 'pos'. We average the representations of each of these nodes to obtain a graph embedding.

4. `GCN`: unlike the previous approaches where a graph embedding is generated before the training phase, we can directly feed the graph data into a GCN and train it end-to-end, thus allowing the network to learn a task-specific graph representation. We base our model on GraphSAGE-GCN [4], using an architecture based on this model from the PyTorch Geometric library[3] that we modify to account for additional node features (tags, gazetteers classes, etc). This allows the network to learn a graph representation that is specific to this task.

### 3.3   Results

In Table 1, we observe a significant decrease in performance for all models between the evaluation and test sets (with a varying intensity depending on the choice of the model) that is probably due to the fact that the test set contains a lot of out-of-vocabulary words that do not appear in the training set. Thus, they lack a node representation that we can feed to the network in inference time. We also see that adding the external knowledge consistently improve the performance of the graph models on both Micro-F1 and Macro-F1 for all models considered. Finally, while the performance on the test set for all graph-only models is still behind LUKE, the best performing state of the art NER model on ConLL 2003, we observe that these models are significantly smaller and thus faster to train (in matters of minutes once the graph embeddings are generated), when using a simple 2-layers feed-forward neural as a classifier. These preliminary results show promising directions for additional investigations and improvements.

## 4   Conclusion and Future work

While the method proposed in this paper shows some promising results, the performance on the ConLL 2003 test set is still significantly lower than the

---

[3] https://github.com/rusty1s/pytorch_geometric/blob/master/examples/proteins_topk_pool.py

| Method | Dev m-F1 | Dev M-F1 | Test m-F1 | Test M-F1 |
|---|---|---|---|---|
| Auto-encoder | 91.0 | 67.3 | 90.3 | 63.2 |
| Auto-encoder+ | 91.5 | 71.7 | 91.5 | 70.4 |
| Node2Vec | 93.3 | 81.6 | 90.0 | 68.3 |
| Node2Vec+ | 94.1 | 82.1 | 91.1 | 72.6 |
| TransE | 91.8 | 75.0 | 91.7 | 70.0 |
| TransE+ | 93.6 | 78.8 | 91.9 | 74.5 |
| GCN | 96.1 | 86.3 | 92.9 | 78.8 |
| GCN+ | 96.5 | 88.8 | 94.1 | 81.0 |
| LUKE [5] | | | | **94.3** |

**Table 1.** NER results with different graph representations (CoNLL-2003 dev and test sets). The entries marked with "+" represent the models with external knowledge added to the words and their context.

best state-of-the-art Transformer-based method as of today. However, we have made multiple design choices to limit the models search space and we believe that additional work on the models themselves (different architectures, hyperparameters fine-tuning, adding attention, changing the classifier) can improve the results. The drop of performance from the validation to the test set is probably due to the lack of any external linguistic knowledge outside of the training set, which can be overcome by enriching the nodes with linguistic features such as Word Embeddings. We will further study the gain from each of the added external knowledge, and test the method on other specialized datasets in order to demonstrate its value for domain-specific applications (fine-grained entity typing). To facilitate reproducibility, we published the code of our experiments at `https://github.com/D2KLab/GraphNER`.

## Acknowledgments

This work has been partially supported by the French National Research Agency (ANR) within the ASRAEL (ANR-15-CE23-0018) and ANTRACT (ANR-17-CE38-0010) projects, and by the European Union's Horizon 2020 research and innovation program within the MeMAD (GA 780069) project.

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
