# OpenReview forum: "Named Entity Recognition as Graph Classification"
_eswc-conferences.org/ESWC/2021/Conference/Poster_and_Demo_Track — ESWC2021 P&D_

### Official Review · AnonReviewer1 · 2021-04-07
**Original but needs improvements**

**Rating:** 4
**Confidence:** 3

**Review:**

This poster paper presents a novel idea for named entity recognition. Existing approaches consider the sentence input as a sequence of words, while the authors want to consider each word as a graph. In a word’s graph, every word from the context is a node, and the central word is linked to his context’s words and to features (like POS tag, gazetteers, etc.). While the idea is original, experiments are still at a preliminary phase.

Pros
-	Common problem but new approach
-	Figure 1 helps to understand the approach
-	Some preliminary experiments

Cons
-	The graph needs to be transformed into a fixed-length vector. This vector is supposed to represent the word, its context, and several features (e.g., POS tag, case, etc.). We can guess that some words will have a lot of features whereas some will not. For instance, gazetteers mentions can be empty or very populated. Considering this heterogeneity, it is hard to visualize a good way to embed this information into a vector. In the early experiments, 4 existing methods to get representations are used, but they are not explained. This part needs to be detailed.
-	In the experiments, the authors use a training set. For each word of this set, they construct the graph, and transform the graph into a vector. Then, it is not very clear. The last sentences of section 3 seem to imply that these vectors are used as input in a multi-layer perceptron. But this needs to be said clearly and explicitly in Section 2, as it is at the core of the approach.
-	Results are not very convincing. We can see that adding external knowledge in a graph representation helps the classifier to perform better. But the best graph representation is still below the best existing classifier that is not based on graphs (LUKE). The fact that a very simple ML classifier is used can give us hope for better results with other ML classifiers, but I think that at least some other ML classifiers should have been tested to see if they can be closer to LUKE, or even above.

Remarks:
-	MLP is mentioned without any information about what it stands for.
-	In table 1, I do not see the interest of the left side (dev accuracy).
-	I do not understand the sentences about the “out-of-vocabulary words that do not appear in the training set” and the “lack of any external linguistic knowledge outside of the training set”. I mean, even if a word is not in the training set, you are able to make the graph representation in the same way as for the training set (using a POS tagger, a chunker, querying wikidata, etc.), and then to represent it as a vector, are not you?

In my opinion, experiments could be reinforced with other classifiers but are acceptable for a poster paper. Nevertheless, the paper needs to be clarified on several points, that is why I reject the paper.

**Anonymity:**

Yes, I would like my review to remain anonymous.

---

### Official Review · AnonReviewer2 · 2021-04-11
**Review of Paper28**

**Rating:** 5
**Confidence:** 3

**Review:**

Quality: good

Clarity: fair

Originality: fair

Significance of this work: fair

Pros:
1.	This paper provides one alternative way to embed information (e.g., gazetteer mentions, morphological form, and linguistic tags) for NER, and demonstrates its effectiveness.
2.	This work contributes a new dataset for NER.
3.	The source code and dataset of the experiment can be assessed.

Cons:
1.	Some results of LUKE in Table 1 is lost.
2.	Lack the detailed analyzes of properties related to NER task in evaluation.
3.	The originality of this work is not enough.


**Anonymity:**

Yes, I would like my review to remain anonymous.

---

### Official Review · ~Vassil_Momtchev1 · 2021-04-14
**Accepted submission and relevance to ESWC Posters and Demos Track**

**Rating:** 8
**Confidence:** 4

**Review:**

The submission of "Named Entity Recognition as Graph Classification" complies with the "Posters and Demos Track" guidelines. Overall the paper is well written, clear to follow, and easy to understand. The Github project is a good starting point to reproduce the results, although some resource files seem to be missing.

The paper presents a novel approach for NER using a classification of graph embeddings, which allows encoding not only contextual information but also external KB data. Although the initial paper results are significantly lower than the state of the art, I enjoy the proposed approach because it enables the easy extension of the model with external knowledge. The authors openly share a significant design limitation of the approach - the lack of external linguistic knowledge, which causes a substantial drop of the F1 between dev and test.

As a reader, it is a bit unclear to me how much/if the structure of the graph helps to improve the result, compared to using the same algorithms with flat lists of values e.g. testing the negative hypothesis.

Here are few comments after reading the paper:

1. "how one can inject new (and specialized) information into these models in a way that it does not require retraining them from scratch" - BERTS, similar to other models, allows you to retrain the last X layers
2. "how to retrain them when big volumes of data are not available" - it is unclear if this approach claims somehow faster learning or a general challenge with all ML-based methods
3. Fig 1 - although the general model is clear, I lacked a concrete example before looking into the code what is the representation of the actual graph with all other features like POS, etc; the knowledge embeddings dimensions are even less obvious for an external reader
4. I suspect some of Github's resource files (i.e. conceptnet_en.csv) are missing due to their size

Edited: Based on the other comments I decrease the scope to clear accept. Still, I would argue that many of the answers raised below are answered if you read the Github project, though from a research point of view I understand why somebody may be frustrated if they don't want to check the Jupiter notebooks.

**Anonymity:**

No, I would like my review to be deanonymized.

---

### Official Review · Program_Chairs · 2021-04-18
**Metareview: Accept (But requires adding further details and clarifying limitations)**

**Rating:** 6
**Confidence:** 5

**Review:**

This was a somewhat controversial paper, with both positive and negative reviews. In terms of the negative reviews, there are certain key details that are lacking, and there are question marks about certain technical design choices that the authors have made. Overall, we have decided to accept the paper with the requirement that the authors add the missing details requested by the reviewers. Regarding some of the technical issues, the P&D track is traditionally a good venue to discuss ongoing work, but we ask the authors to clearly clarify limitations and to outline next steps to improve the work (e.g., based on the comments of the reviewers).

**Anonymity:**

Yes, I would like my review to remain anonymous.

---

### Decision · Program_Chairs · 2021-04-19

Accept